# Homeostatic regulation of STING by retrograde membrane traffic to the ER

Kojiro Mukai [1], Emari Ogawa[2], Rei Uematsu[2], Yoshihiko Kuchitsu[1], Fumika Kiku[2], Takefumi Uemura[3], Satoshi Waguri [3], Takehiro Suzuki[4], Naoshi Dohmae[4], Hiroyuki Arai[2], Anthony K. Shum [5] & Tomohiko Taguchi [1]✉

Coat protein complex I (COP-I) mediates the retrograde transport from the Golgi apparatus to the endoplasmic reticulum (ER). Mutation of the *COPA* gene, encoding one of the COP-I subunits (α-COP), causes an immune dysregulatory disease known as COPA syndrome. The molecular mechanism by which the impaired retrograde transport results in autoinflammation remains poorly understood. Here we report that STING, an innate immunity protein, is a cargo of the retrograde membrane transport. In the presence of the disease-causative α-COP variants, STING cannot be retrieved back to the ER from the Golgi. The forced Golgi residency of STING results in the cGAS-independent and palmitoylation-dependent activation of the STING downstream signaling pathway. Surf4, a protein that circulates between the ER/ ER-Golgi intermediate compartment/ Golgi, binds STING and α-COP, and mediates the retrograde transport of STING to the ER. The STING/Surf4/α-COP complex is disrupted in the presence of the disease-causative α-COP variant. We also find that the STING ligand cGAMP impairs the formation of the STING/Surf4/α-COP complex. Our results suggest a homeostatic regulation of STING at the resting state by retrograde membrane traffic and provide insights into the pathogenesis of COPA syndrome.

[1] Laboratory of Organelle Pathophysiology, Department of Integrative Life Sciences, Graduate School of Life Sciences, Tohoku University, Sendai, Japan. [2] Department of Health Chemistry, Graduate School of Pharmaceutical Sciences, University of Tokyo, Tokyo, Japan. [3] Department of Anatomy and Histology, Fukushima Medical University School of Medicine, Fukushima, Japan. [4] Biomolecular Characterization Unit, RIKEN Center for Sustainable Resource Science, Wako, Japan. [5] Department of Medicine, Division of Pulmonary and Critical Care, University of California San Francisco, San Francisco, CA, USA. ✉email: tom_taguchi@tohoku.ac.jp

The COPA syndrome is a recently discovered monogenic disorder of immune dysregulation characterized by high-titer autoantibodies, interstitial lung disease, inflammatory arthritis, and high expression of type I interferon-stimulated genes[1,2]. The disease is caused by heterozygous mutations of the *COPA* gene, encoding the α subunit (α-COP) of COP-I that mediates the retrograde transport of proteins from the Golgi to the endoplasmic reticulum (ER)[3,4]. All of the mutations[1] of the disease-causative α-COPs lie in the N-terminal WD40 domain (Supplementary Fig. 1a), which has been implicated in the recognition of cargo proteins[5]. How the retrograde transport in the COPA syndrome causes the immune dysregulatory disease remains largely unknown.

Vertebrates have evolved biological systems to combat invading pathogens. As the first line of host defense, the innate immune system detects microbial pathogens with pattern recognition receptors (PRRs) that bind unique pathogen-associated molecular patterns (PAMPs)[6,7]. Activated PRRs initiates intracellular signaling cascades, leading to the transcriptional expression of proinflammatory cytokines, type I interferons, and other antiviral proteins that all coordinate the elimination of pathogens and infected cells. Viral RNA, cytosolic DNA, or the gram-negative bacterial cell-wall component lipopolysaccharide serves as PAMP that activates a distinct signaling pathway, such as RIG-I/MAVS, cGAS/STING, or TLR4/TRIF pathway. MAVS, STING, or TRIF activates the downstream protein kinase TBK1, which then phosphorylates and activates interferon regulatory factor 3 (IRF3), the essential transcription factor that drives type I interferon production[8].

STING[9] is an ER-localized transmembrane protein. After STING binding to cyclic dinucleotides (CDNs)[10] that are generated by cGAMP synthase (cGAS)[11], an enzyme that is activated by the presence of cytosolic DNA, STING translocates to the Golgi where STING activates TBK1 at the trans-Golgi network (TGN)[12–14]. Because α-COP is a component of COP-I that mediates the membrane transport between the Golgi and the ER, we reasoned that the disease-causative α-COP variant (K230N, R233H, E241K, or D243G; the α-COP variant hereafter)[1] could influence the STING pathway.

In this work, we show that the disease-causing COPA variants prevent STING transport to the ER, leading to cGAS-independent activation of the STING pathway.

## Results

### The α-COP variants activate the STING pathway

We performed luciferase assay with HEK293T cells that lack endogenous STING. After co-transfection with α-COP, STING, and a luciferase reporter construct with IRF3 (also known as ISRE or PRD III-I)-responsive promoter elements, the luciferase activity in the total cell lysate was measured. Wild-type α-COP did not activate the IRF3 promoter regardless of the expression of STING, while all the α-COP variants activated the IRF3 promoter in STING-expressing cells (Fig. 1a). The α-COP variants did not activate the IRF3 promoter in cells transfected with MAVS or TRIF (Supplementary Fig. 1b, c).

The effects of the α-COP variants on STING signaling were modest in this transient transfection system, possibly because of STING overexpression[10]. Therefore, we examined the effects of wild-type or mutant α-COP in cells stably expressing STING in *Sting*[−/−] MEFs. The amount of STING in *Sting*[−/−] MEFs stably expressing EGFP-STING was ~3-fold higher than that of endogenous STING in WT MEFs (Supplementary Fig. 1d). We then stably expressed wild-type or mutant α-COP in *Sting*[−/−] MEFs expressing EGFP-STING. Both wild-type and mutant α-COP localized mainly at the Golgi (Supplementary Fig. 2a–e).

Other COP-I subunits (β, β′, δ, and γ) also localized at the Golgi in these cells (Supplementary Fig. 2a–d). We also examined the expression levels of these COP-I subunits by western blot, and found that the stable expression of α-COP did not affect that of other COP-I subunits (Supplementary Fig. 2f, g). We then examined activation of STING pathway in these cells. Phosphorylated TBK1 (p-TBK1), a hallmark of STING activation, emerged only in cells expressing the α-COP variants (Fig. 1b and Supplementary Fig. 18a). Several innate immunity genes, which are reported to be upregulated in MEFs expressing constitutively active STING[15], were also upregulated in cells expressing the α-COP variants (Fig. 1c and Supplementary Fig. 3). These results suggested that expression of the α-COP variants could activate the STING pathway. Given that the α-COP variants can be defective in cargo recognition[1] (Supplementary Fig. 1a) and that the localization and amount of COP-I subunits including endogenous α-COP were not affected by the expression of the α-COP variants (Supplementary Fig. 2), the α-COP variants would decrease the number of functional COP-I complex. Therefore, the effect of the disease-causative α-COP variants on activation of STING pathway may be dominant negative.

### The α-COP variants alter STING localization to the Golgi

We next examined the subcellular localization of STING. In cells expressing wild-type α-COP, EGFP-STING distributed throughout the cytoplasm and co-localized with calreticulin (an ER protein), indicating that STING localized at the ER (Fig. 2a and Supplementary Figs. 4 and 18b). p-TBK1 and phosphorylated STING at Ser365 (p-STING) that is generated by active TBK1[8,16] were not detected. In contrast, in cells expressing the α-COP variants, EGFP-STING mostly localized at perinuclear compartments that include the Golgi (Fig. 2a and Supplementary Figs. 4–7 and 18b). Thus, the expression of the α-COP variants altered the STING localization. The signals of p-TBK1 and p-STING emerged in these cells (Fig. 2b and Supplementary Figs. 8–10 and 18c), being consistent with the activation of STING (Fig. 1). Immunoelectron microscopy corroborated the Golgi localization of STING in cells expressing the α-COP variant (E241K; Fig. 2c and Supplementary Figs. 18d and 19). Given that COP-I mediates the retrograde transport from the Golgi to the ER[4], these results suggested that STING is a cargo of the retrograde transport of COP-I and that STING cannot be retrieved back to the ER in the presence of the disease-causative α-COP variants. The amount of STING that was co-immunoprecipitated with the α-COP variants was smaller than that with wild-type α-COP (Fig. 2d), further supporting this notion.

### Surf4 binds STING and α-COP, and is required for STING localization to the ER

α-COP binds *C*-terminal di-lysine motifs of its cargo proteins, such as KKXX and KXKXX[3,17,18]. As STING does not possess these motifs at its *C*-terminus, we reasoned the presence of adapter protein(s) that mediates the interaction of STING and α-COP. We analyzed STING-binding proteins by mass spectrometry and 18 proteins with these motifs were identified (Supplementary Table 1). We knock-downed these proteins individually with siRNAs and examined the effect on the subcellular localization of STING. We found that knockdown of Surf4, not that of the other 17 proteins, altered the localization of STING to the Golgi (Fig. 3a, b and Supplementary Figs. 11 and 12). Knockdown of Surf4 also resulted in the emergence of p-TBK1 in *Sting*[−/−] MEFs that were reconstituted of EGFP-STING, but not in *Sting*[−/−] MEFs (Fig. 3c). These results suggested that Surf4 is critical to maintain the steady-state localization of STING to the ER.

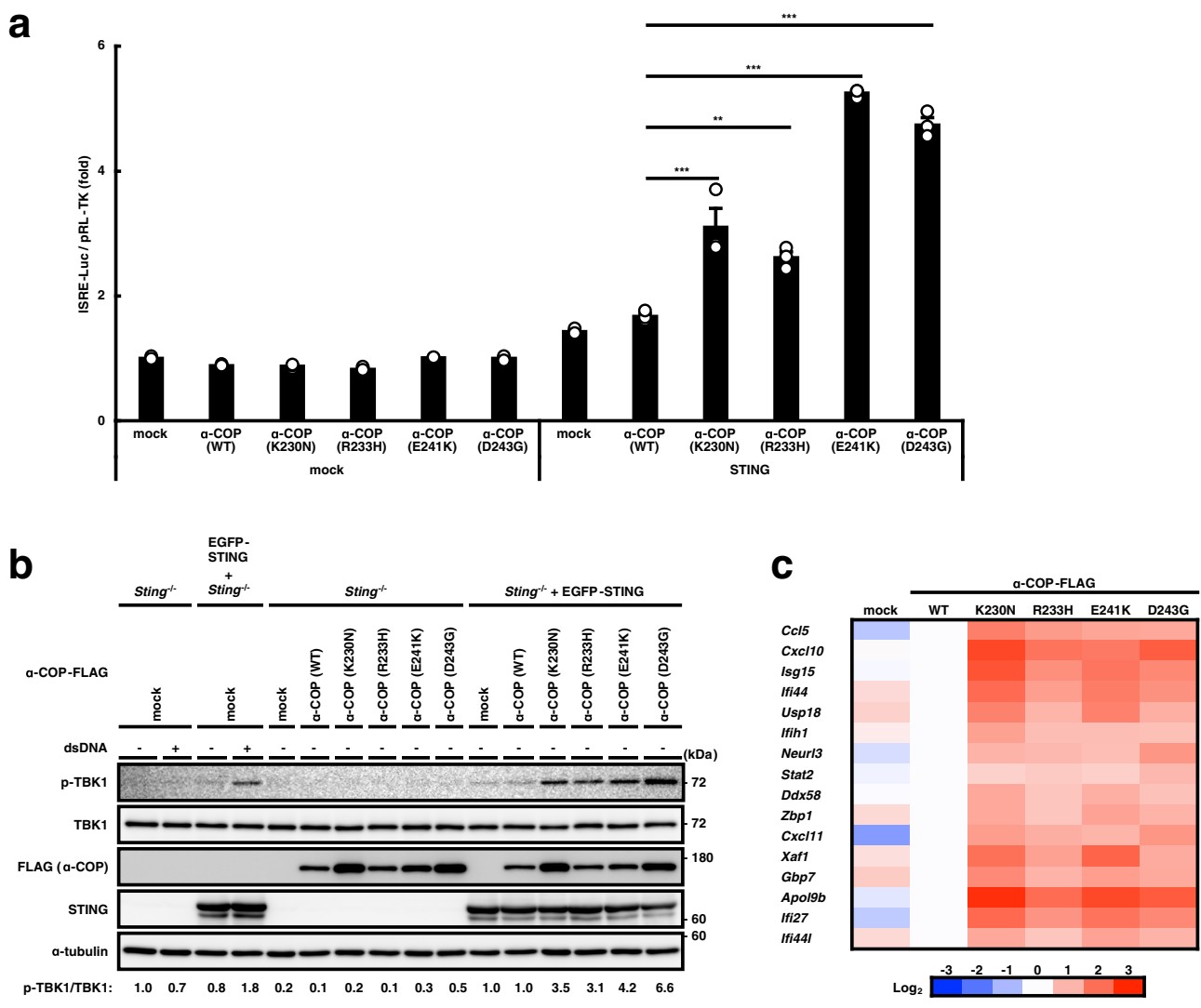

**Fig. 1 The α-COP variants activate the STING pathway. a** HEK293T cells were transfected as indicated, together with an ISRE (also known as PRDIII or IRF-E)-luciferase reporter. Luciferase activity was then measured. Data represent mean s.e.m. of three independent experiments. **b** STING and/or α-COP were stably expressed in $Sting^{-/-}$ MEFs. Cell lysates were prepared and analyzed by western blot. **c** Quantitative real-time PCR (qRT-PCR) of the expression of innate immune genes in MEFs expressing the α-COP variants. The indicated gene expression was normalized on the basis of GAPDH content and the $\log_2$ fold change compared to α-COP (WT) was plotted as a heatmap. See also Supplementary Fig. 3.

Surf4, a multi-pass transmembrane protein, cycles among the ER, ERGIC, and the Golgi. Surf4 functions in the anterograde trafficking pathway from the ER[19]. Surf4 is involved in membrane recruitment of COP-I[20], suggesting that Surf4 also functions in the retrograde trafficking pathway. Indeed, we found the interaction between Surf4 and wild-type α-COP by co-immunoprecipitation analysis (Fig. 3d). The mutant Surf4 (K265A/K266A/K267A), in which the C-terminal lysine residues were substituted to alanine, exhibited a reduced binding to α-COP, suggesting that the interaction between Surf4 and COPA is, at least in part, mediated through the C-terminal consecutive lysine residues on Surf4 (Fig. 3d and Supplementary Fig. 18e). We performed the knockdown/rescue experiments with wild-type and the mutant Surf4. The expression of wild-type Surf4, but not the mutant Surf4 (K265A/K266A/K267A), rescued the ER localization of STING (Supplementary Fig. 13c). These results suggested that the interaction of Surf4 with α-COP through the C-terminal KKK motif (Fig. 3d) was essential for the STING retrieval to the ER. The disease-causative α-COP variant (E241K) exhibited a reduced binding to Surf4 (Fig. 3e and Supplementary

Fig. 18e). Moreover, knockdown of Surf4 reduced the binding between STING and α-COP (Supplementary Fig. 13d). These results suggested that Surf4 serves as a cargo receptor for STING in the retrograde transport mediated by COP-I.

We examined the subcellular localization of Surf4. Surf4 localized at punctate structures, some of which were positive with α-COP, but not with TGN38 (Supplementary Fig. 14a), suggesting that these puncta represent ERGIC. Surf4 co-localized with ERGIC53 (a protein that circulates among the ER, ERGIC, and the Golgi) at these punctate structures (Supplementary Fig. 14b), supporting that these puncta indeed represent ERGIC. Based on these results, in particular the one that Surf4 co-localized with α-COP in ERGIC, we assume that Surf4 was involved in the retrograde transport of STING from the ERGIC to the ER.

**STING activation with the α-COP variants requires the retrograde membrane traffic and palmitoylation of STING, but not cGAS.** Activation of the STING signaling pathway with cGAMP requires the ER-to-Golgi traffic of STING and

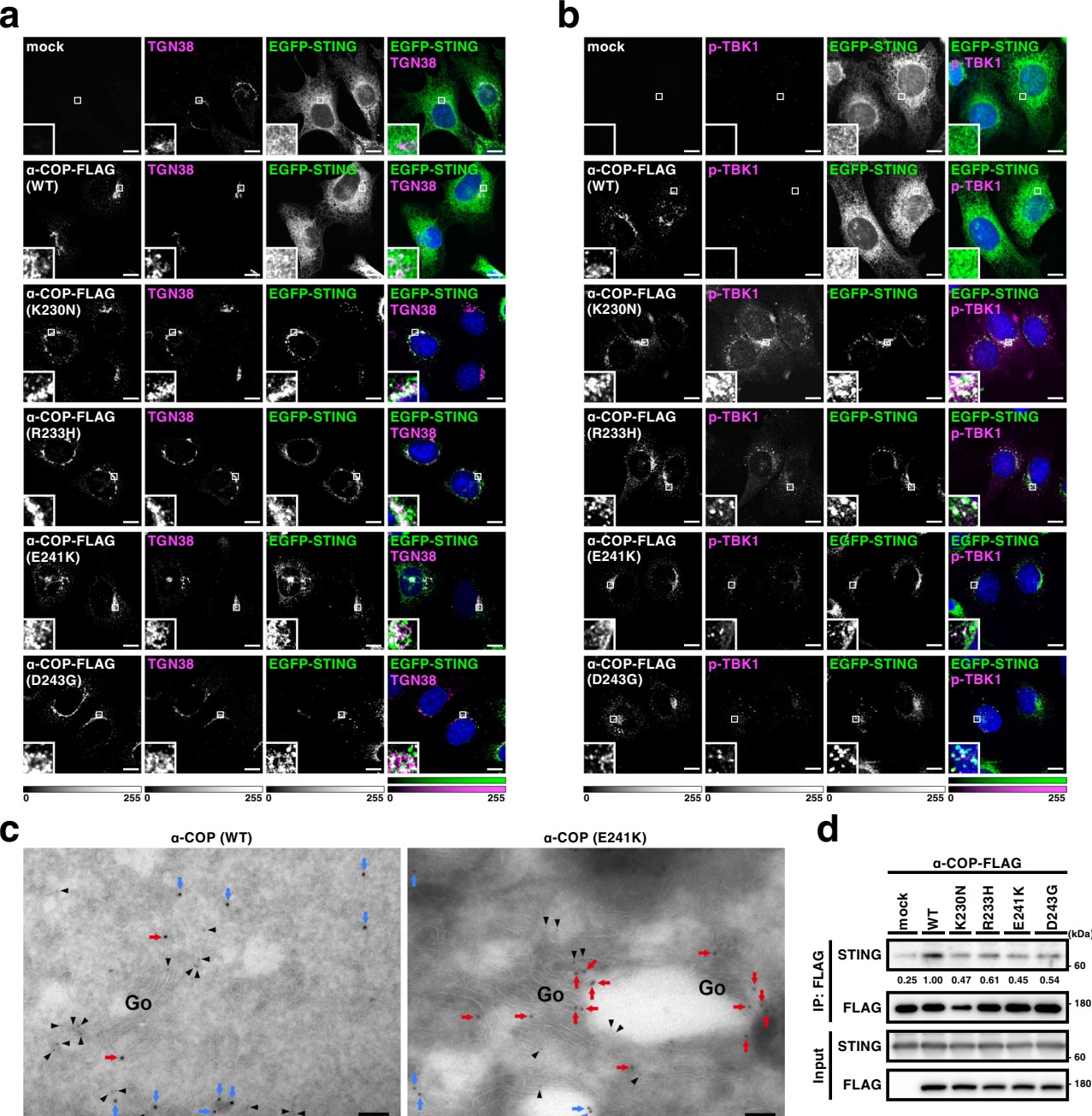

**Fig. 2 The α-COP variants alter STING localization to the Golgi. a**, **b** α-COP and EGFP-STING were stably expressed in *Sting*[−/−] MEFs as indicated. Cells were fixed, permeabilized, and stained for TGN38 (a Golgi protein) (**a**) or p-TBK1 (**b**). Nuclei were stained with DAPI (blue). Color scales and intensity levels are indicated below images. Scale bars, 10 μm. **c** α-COP-FLAG (WT) or α-COP-FLAG (E241K) was stably expressed in EGFP-STING-reconstituted MEFs. Cells were fixed and processed for ultrathin cryosections. They were immunostained with anti-GFP (rabbit) and anti-FLAG (mouse) antibodies. As secondary antibodies, colloidal gold particle-conjugated donkey anti-rabbit antibody (12 nm) and anti-mouse (6 nm) were used. Arrowheads indicate α-COP signal, red arrows indicate GFP signal on the Golgi, and blue arrows indicate GFP signal that was not associated with the Golgi. Go, the Golgi stack. Scale bars, 500 nm. **d** Cell lysates were prepared from MEFs expressing various α-COP as indicated, and α-COP was immunoprecipitated with anti-FLAG antibody. Cell lysates and the immunoprecipitates were analyzed by western blot.

palmitoylation of STING at the Golgi[12,13,21]. We treated cells expressing the α-COP variants with brefeldin A (BFA), an agent to block the ER-to-Golgi traffic[22], or two palmitoylation inhibitors [a pan-palmitoylation inhibitor 2-bromopalmitate (2-BP) and a mouse STING-specific palmitoylation inhibitor C-178[23]] and found that these treatments suppressed phosphorylation of TBK1 and the expression of Ccl5, Cxcl10, and Apol9b (Fig. 4a, b and Supplementary Fig. 15c). These results suggested that STING

activation with the α-COP variants, as with cGAMP, requires the ER-to-Golgi traffic and palmitoylation of STING.

We asked if the activation of STING caused by impaired retrograde transport requires cGAMP. To address this question, we prepared cGAS-knockout MEFs by CRISPR-Cas9 system (Supplementary Fig. 16a). In cGAS-knockout MEFs expressing the α-COP variants, (i) p-TBK1 still emerged (Fig. 4c), (ii) the gene expression of Ccl5, Cxcl10, and Apol9b were induced

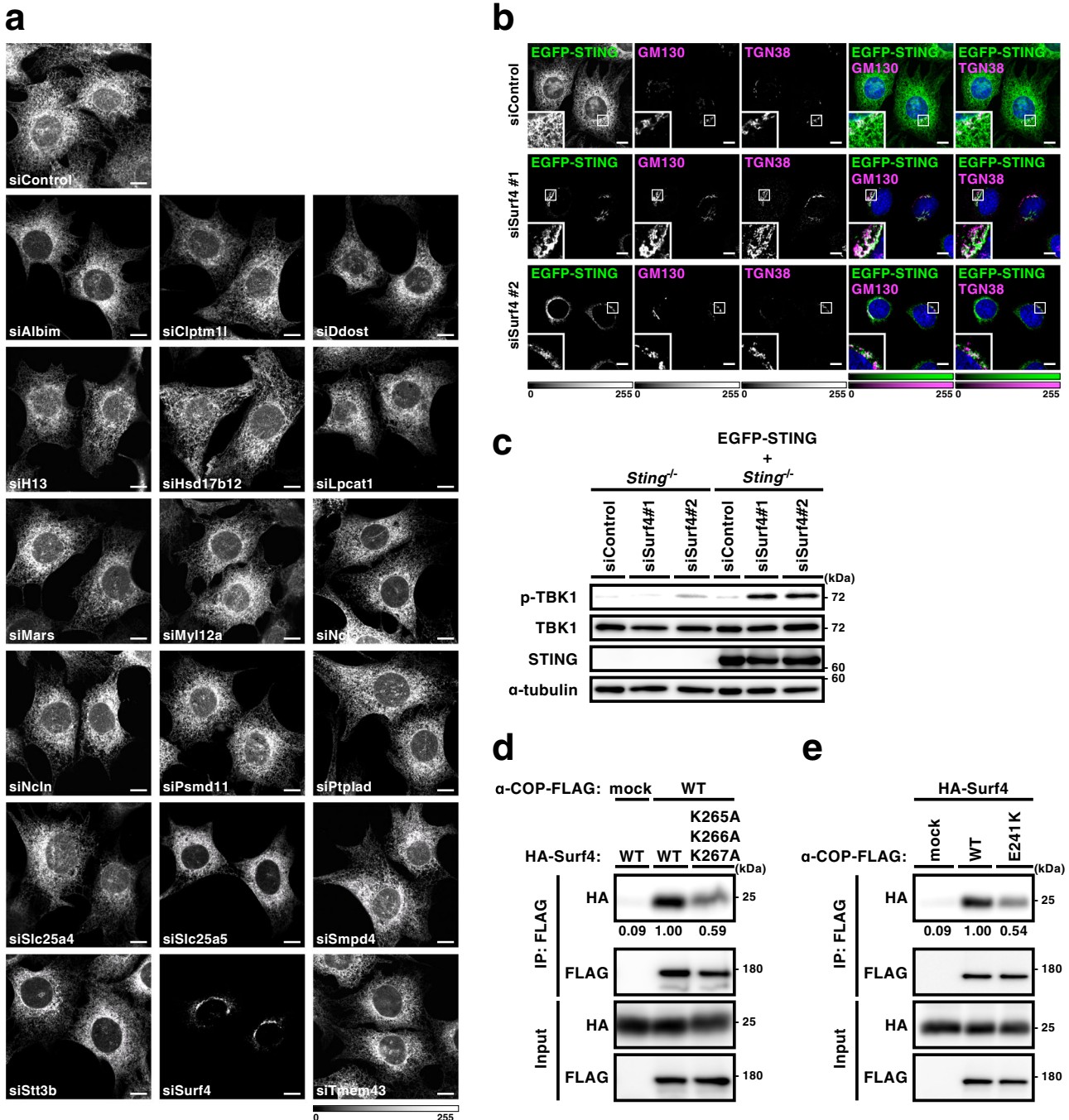

**Fig. 3 Surf4 binds STING and α-COP, and is required for STING localization to the ER. a, b** MEFs expressing EGFP-STING were treated with the indicated siRNA for 48 h. Cells were fixed, permeabilized, and stained for GM130 and TGN38. Color scales and intensity levels are indicated below images. Scale bars, 10 μm. **c** Sting$^{-/-}$ MEFs or Sting$^{-/-}$ MEFs reconstituted of EGFP-STING were treated with the indicated siRNA. Cell lysates were prepared and analyzed by western blot. **d, e** HEK293T cells were transfected with the indicated plasmids. Cell lysates were prepared and α-COP-FLAG was immunoprecipitated. Cell lysates and the immunoprecipitates were analyzed by western blot.

(Supplementary Fig. 16b), and (iii) STING translocated to perinuclear compartments that include the Golgi (Supplementary Fig. 16c). We depleted Surf4 by siRNA in cGAS-knockout cells and found that STING still translocated to the Golgi (Supplementary Fig. 13a) and that TBK1 was phosphorylated (Supplementary Fig. 13b). These results suggested that the translocation of STING from the ER to the Golgi does not necessarily require cGAMP and that the forced Golgi residency of STING with the impaired retrograde transport would suffice to activate the STING signaling pathway in the absence of cGAMP.

The translocation of STING from the ER to the Golgi in cGAS-knockout MEFs expressing the α-COP variants were suppressed by knockdown of Sar1a/b (Supplementary Fig. 17), which are two small GTPases responsible for the ER exit of STING[13]. These results led us to propose a model to explain how the membrane traffic axis between the ER and the Golgi is integrated into the STING signaling pathway (Fig. 4d). STING exits the ER without cGAMP by COP-II-mediated anterograde transport (Supplementary Fig. 17). This exit may be with bulk flow as part of the membrane without association with COP-II coat subunits. Once

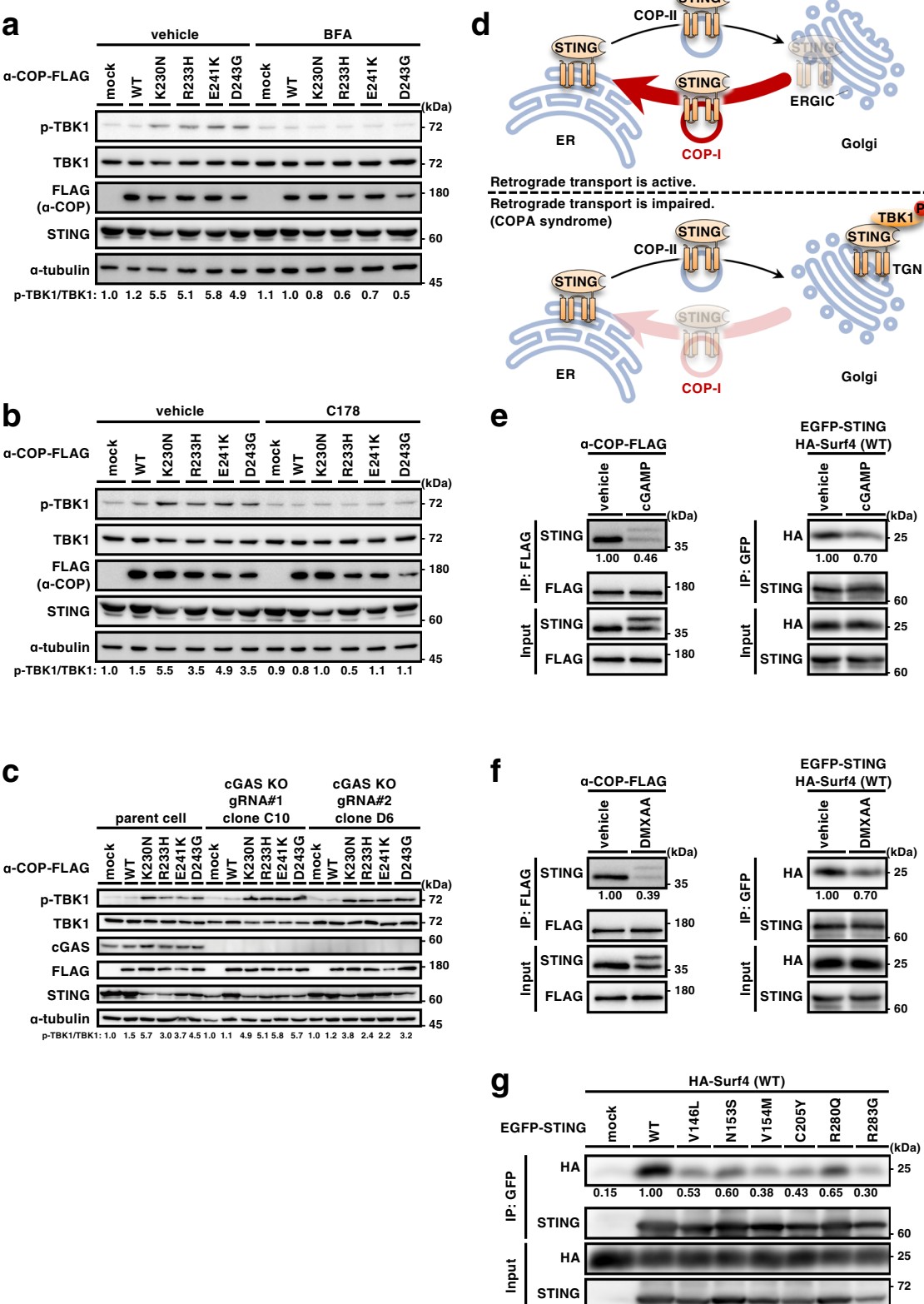

**Fig. 4 STING activation with the α-COP variants requires the ER-to-Golgi traffic and palmitoylation of STING, but not cGAS. a**, **b** MEFs expressing various α-COP as indicated were treated with BFA (0.3 μg/ml) for 8 h (**a**) or with C178 (10 μM) for 8 h (**b**). Cell lysates were prepared and analyzed by western blot. **c** α-COP was stably expressed in cGAS-KO MEFs. Cell lysates were prepared and analyzed by western blot. **d** A model of STING regulation by membrane traffic between the ER and the Golgi. **e**, **f** α-COP-FLAG was stably expressed WT MEFs, or HA-Surf4 was stably expressed in *Sting^-/-* MEFs that were reconstituted of EGFP-STING. Cells were stimulated with cGAMP (**e**) or DMXAA (**f**) for 1 h. Cell lysates were prepared, and α-COP-FLAG or EGFP-STING was immunoprecipitated. Cell lysates and the immunoprecipitates were analyzed by western blot. **g** HEK293T cells were transfected with the indicated plasmids. Cell lysates were prepared and EGFP-STING were immunoprecipitated. Cell lysates and the immunoprecipitates were analyzed by western blot.

STING reaches the ERGIC (Supplementary Fig. 14), STING is retrieved back to the ER by the COP-I-mediated retrograde transport. In a condition where the retrograde transport is impaired, such as in the presence of the disease-causative α-COP variants (Fig. 2), or where retrograde pathway is saturated because of STING overexpression[9,10], STING is forced to accumulate at the Golgi. STING is then subjected to palmitoylation and activates TBK1 at TGN[12] (Supplementary Fig. 8).

Intriguingly, STING with DMXAA (a mouse STING agonist) or with cGAMP exhibited a reduced binding to α-COP and Surf4 (Fig. 4e, f). Therefore, the translocation of STING from the ER and activation of the STING signaling pathway with cGAMP may be partly due to the reduced ability of the STING/ligand complex to be packaged into COP-I transport vesicles. Mutations in STING are found in patients with an autoinflammatory disease called STING-associated vasculopathy with onset in infancy (SAVI) and these mutations appear to make STING constitutively active[24–26]. The SAVI variants exit the ER without cGAMP[13] and require the ER-to-Golgi traffic and palmitoylation for their activity[12,13]. We found that all the six SAVI variants exhibited a reduced binding to Surf4 (Fig. 4g). The reduced binding may result in the impaired retrograde transport of STING to the ER, which would partly explain the aberrant localization of the SAVI variants to the Golgi without cGAMP.

## Discussion

In this study, we demonstrate the homeostatic regulation of STING at the resting state by the retrograde membrane traffic. Intriguingly, this finding is corroborated in a recently described mouse model of COPA syndrome ($Copa^{E241K/+}$ mice)[27]: $Copa^{E241K/+}$ mice exhibit spontaneous activation of STING with upregulation of type I interferon signaling and systemic inflammation, all of which is abrogated in STING-deficient animals[28]. However, given the multiple cargo proteins transported by COP-I vesicles, other effects of α-COP loss-of-function may also contribute to the COPA syndrome-specific symptoms, which are not manifested in SAVI in which STING is constitutively active because of the STING mutations. Nonetheless, our results that the inflammatory response in the presence of the α-COP variants can be effectively suppressed by STING palmitoylation inhibitors may provide a treatment approach for COPA syndrome patients.

## Methods

**Antibodies**. Antibodies used in this study were as follows: mouse anti-GFP (JL-8, dilution 1:1000; Clontech); rabbit anti-β-COP (PA1-061, dilution 1:2000 for western blot and immunofluorescence), rabbit anti-mCherry for detecting mScarlet-I (PA5-34974, dilution 1:1000 for western blot), and Alexa 488-, 594-, or 647-conjugated secondary antibodies (A21202, A21203, A21206, A21207, A31573, A11016, A21448, dilution 1:2000; Thermo Fisher Scientific); rabbit anti-TBK1 (ab40676, dilution 1:1000; Abcam); rabbit anti-phospho-TBK1 (D52C2, dilution 1:1000 for western blot, dilution 1:100 for immunofluorescence), rabbit anti-cGAS (D3O8O, dilution 1:1000), and rabbit anti-phospho-STING (D1C4T, dilution 1:400; Cell signaling); mouse anti-calreticulin (612136, dilution 1:1000), and mouse anti-GM130 (610823, dilution 1:1000; BD Biosciences); rabbit anti-α-COP (HPA028024, dilution 1:1000 for western blot), mouse anti-α-tubulin (DM1A, dilution 1:5000) and mouse anti-FLAG M2 antibody (Sigma); Goat Anti-Rabbit IgG(H + L) Mouse/Human ads-HRP (4050-05, dilution 1:10,000) and Goat Anti-Mouse IgG(H + L) Human ads-HRP (1031-05, dilution 1:10,000; Southern Biotech); sheep anti-TGN38 (AHR499G, dilution 1:500) (Serotec); rabbit anti-STING antibody (19851-1-AP, dilution 1:1000 for western blot), rabbit anti-γ-COP (12393-I-AP, dilution 1:1000 for western blot, dilution 1:200 for immunofluorescence), and rabbit anti-ERGIC53 (13364-1-AP, dilution 1:1000 for western blot, dilution 1:200 for immunofluorescence; Proteintech); mouse anti-HA (4B2, dilution 1:1000 for western blot and immunofluorescence) and mouse anti-FLAG (1E6, dilution 1:1000 for western blot and immunofluorescence; dilution 1:50 for immunoelectron microscopy; Wako); rabbit anti-β′-COP (A304-523A-T, dilution 1:1000 for western blot, dilution 1:200 for immunofluorescence; Bethyl Laboratories); mouse anti-δ-COP (GTX630562, dilution 1:1000 for western blot, dilution 1:200 for immunofluorescence; GeneTex); 12 nm colloidal gold particle-conjugated donkey anti-rabbit antibody (711-205-152, dilution 1:20) and 6 nm colloidal gold particle-conjugated donkey anti-mouse IgG (715-195-150, dilution 1:20; Jackson ImmunoResearch laboratories). For the immunoprecipitation of FLAG-tagged

protein, anti-FLAG M2 Affinity Gel (A2220, Sigma) was used. For the immunoprecipitation of GFP-tagged protein, anti-GFP nanobody was used. pGEX6P1-GFP-Nanobody was a gift from Kazuhisa Nakayama (Addgene plasmid # 61838).

**Reagents**. The following reagents were purchased from the manufacturers as noted: BFA (Sigma); 2-BP (Wako). ISD (90-mer), used as dsDNA in this study, was prepared as follows: equimolar amounts of oligonucleotides (sense: 5′-TACAGA TCTACTAGTGATCTATGACTGATCTGTACATGATCTACATACAGATCTAC TAGTGATCTATGACTGATCTGTACATGATCTACA-3′, antisense: 5′-TGTAG ATCATGTACAGATCAGTCATAGATCACTAGTAGATCTGTATGTAGATCA TGTACAGATCAGTCATAGATCACTAGTAGATCTGTA-3′) were annealed in PBS at 70 °C for 30 min before cooling to room temperature. C-178 was provided by Carna Biosciences, Inc.

**Cell culture**. HEK293T cells were purchased from the American Type Culture Collection (ATCC). MEFs were obtained from embryos of WT or $Sting^{-/-}$ mice at E13.5 and immortalized with SV40 Large T antigen. HEK293T and MEFs were cultured in DMEM supplemented with 10% fetal bovine serum/penicillin/streptomycin/glutamine in a 5% $CO_2$ incubator.

MEFs that stably express EGFP-mouse STING or mouse α-COP variants were established using retrovirus. Plat-E cells were transfected with pMX-IP-EGFP-STING or pMX-IB-α-COP-FLAG and the medium that contains the retrovirus was collected. MEFs were incubated with the medium and then selected with puromycin (2 μg/mL) or blasticidin (5 μg/mL) for several days.

**PCR cloning**. Mouse STING was amplified by PCR with complementary DNA (cDNA) derived from ICR mouse liver. The product encoding mouse STING was introduced into pMXs-IPuro–GFP, to generate N-terminal GFP-tagged construct. Mouse α-COP and mouse Surf4 was amplified by polymerase chain reaction (PCR) with cDNA derived from MEFs. The product encoding α-COP was introduced into pMXs-IBla-FLAG, to generate C-terminal FLAG-tagged construct. The product encoding Surf4 was introduced into pMXs-IHyg-HA, to generate N-terminal HA-tagged construct. α-COP variants and Surf4 mutant were generated by site-directed mutagenesis.

**Luciferase assay**. HEK293T cells seeded on 24-well plates were transiently transfected with luciferase reporter plasmid (100 ng), pRL-TK (10 ng) as internal control, STING-expression plasmid in pBabe vector (200 ng), and α-COP-expression plasmid in pMX vector (200 ng) Twenty-four hours after the transfection, the luciferase activity in the total cell lysate was measured.

**qRT-PCR**. Total RNA was extracted from cells using Isogen II (Nippongene), and reverse-transcribed using ReverTraAce qPCR RT Master Mix with gDNA Remover (TOYOBO). Quantitative real-time PCR (qRT-PCR) was performed using KOD SYBR qPCR (TOYOBO) and LightCycler 96 (Roche). The sequences of the primers were provided in Supplementary Table 2. Target gene expression was normalized on the basis of GAPDH content.

**Immunocytochemistry**. Cells were fixed with 4% paraformaldehyde (PFA) in PBS at room temperature for 15 min, permeabilized with 0.1% Triton X-100 in PBS at room temperature for 5 min, and quenched with 50 mM $NH_4Cl$ in PBS at room temperature for 10 min. After blocking with 3% BSA in PBS, cells were incubated with primary antibodies, then with secondary antibodies conjugated with Alexa fluorophore.

**Confocal microscopy**. Confocal microscopy was performed using a LSM880 with Airyscan (Zeiss) with a 63 × 1.4 Plan-Apochromat oil immersion lens or 100 × 1.46 alpha-Plan-Apochromat oil immersion lens. Images were analyzed with Zeiss ZEN 2.3 SP1 FP3 (black, 64 bit) (ver. 14.0.21.201) and Fiji (ver. 2.1.0/1.53c).

**Immunoprecipitation**. Cells were lysed with IP buffer (50 mM HEPES-NaOH (pH 7.2), 150 mM NaCl, 5 mM EDTA, 1% CHAPS, protease inhibitors (Protease Inhibitor Cocktail for Use with Mammalian Cell and Tissue Extracts, 25955-11, nacalai tesque), and phosphatase inhibitors (8 mM NaF, 12 mM beta-glycerophosphate, 1 mM $Na_3VO_4$, 1.2 mM $Na_2MoO_4$, 5 μM cantharidin, and 2 mM imidazole). The lysates were centrifuged at 20,000 × g for 10 min at 4 °C, the resultant supernatants were incubated for overnight at 4 °C with anti-FLAG M2 Affinity Gel or anti-GFP nanobody beads[29] for 1 h. The beads were washed four times with immunoprecipitation wash buffer (50 mM HEPES-NaOH (pH 7.2), 150 mM NaCl, 0.7% CHAPS), and eluted with elution buffer (50 mM HEPES-NaOH (pH 7.2), 150 mM NaCl, 5 mM EDTA, 1% Triton X-100, 200 μg/mL FLAG peptide).

**Immunoelectron microscopy**. Cells were fixed with 4% PFA (1.04005.1000, MERCK), 4% sucrose, and 0.1 M phosphate buffer (pH 7.2) for 10 min at room temperature and then 30 min at 4 °C. After rinsing with 7.5% sucrose and 0.1 M phosphate buffer (pH 7.4), they were scraped and embedded in 10% gelatin (G2500, Sigma) and 0.1 M phosphate buffer (pH 7.4). The cell blocks were cut into

small pieces (~1 mm cube), which were infused overnight with 20% poly-vinylpyrrolidone (PVP10, Sigma-Aldrich), 1.84 M sucrose, 10 mM $Na_2CO_3$, and 0.08 M phosphate buffer (pH 7.4) followed by rapid freezing in liquid nitrogen[30]. Ultrathin cryosections were prepared using an ultramicrotome (EM UC7, Leica) equipped with a cryochamber (EM FC7, Leica). They were incubated with 1% BSA and PBS for 20 min at room temperature, and then with rabbit antibody against GFP (ab6556, dilution 1:100; Abcam) and mouse antibody against FLAG (1E6, dilution 1:50; Wako) for 24 h at 4 °C. After incubation with 12 nm colloidal gold particle-conjugated donkey anti-rabbit (711-205-152, dilution 1:20; Jackson ImmunoResearch) and 6 nm colloidal gold particle-conjugated donkey anti-mouse IgG (715-195-150, dilution 1:20; Jackson ImmunoResearch) for 1 h at room temperature, they were fixed with 2% glutaraldehyde (G017/1, TAAB) and PBS for 5 min. They were stained with 2% uranyl acetate for 5 min and embedded in 0.17% uranyl acetate and 0.33% polyvinyl alcohol (P8136, Sigma-Aldrich). After drying up, sections were observed using an electron microscope (JEM1200EX, JEOL).

**western blot**. Proteins were separated in polyacrylamide gel and then transferred to polyvinylidene difluoride membranes (Millipore). These membranes were incubated with primary antibodies, followed by secondary antibodies conjugated to peroxidase. The proteins were visualized by enhanced chemiluminescence using a LAS-4000 (GE Healthcare) or Fusion SOLO.7S.EDGE (Vilber-Lourmat).

**Mass spectrometry**. Cells were lysed with IP buffer (50 mM HEPES-NaOH (pH 7.2), 150 mM NaCl, 5 mM EDTA, 1% Triton X-100, protease inhibitors, and phosphatase inhibitors). The lysates were centrifuged at $20,000 \times g$ for 10 min at 4 °C, the resultant supernatants were incubated for overnight at 4 °C with anti-FLAG M2 Affinity Gel. The beads were washed four times with immunoprecipitation wash buffer (50 mM HEPES-NaOH (pH 7.2), 150 mM NaCl, and 1% Triton X-100), and eluted with elution buffer (50 mM HEPES-NaOH (pH 7.2), 150 mM NaCl, 5 mM EDTA, 1% Triton X-100, and 500 µg/mL FLAG peptide. Eluted proteins were applied to SDS-PAGE, and the electrophoresis was stopped when the samples were moved to the top of the separation gel. The gel was stained with CBB and the protein bands at the top of separation gel were excised. The proteins were reduced and S-carboxylmethylated, followed by a tryptic digestion in gel (TPCK treated trypsin, Worthington Biochemical Corporation). The digests were separated with a reversed phase nano-spray column (NTCC-360/75-3-105, NIKKYO technos) and then applied to Q Exactiv Hybrid Quadrupole-Orbitrap mass spectrometer (Thermo Scientific). MS and MS/MS data were obtained with TOP10 method. The MS/MS data was searched against NCBI nr database using MASCOT program 2.6 (Matrix Science) and the MS data was quantified using Proteome Discoverer 2.2 (Thermo Scientific).

**RNA interference**. siRNA specific to Ablim1 (M-059643-01), Clptm1l (M-062856-00), Ddost (M-064791-00), H13 (M-059025-01), Hsd17b12 (M-060708-01), Lpcat1 (M-059984-01), Mars (M-066281-00), Myl12a (M-046975-01), Ncl (M-059054-01), Ncln (M-052038-01), Psmd11 (M-057766-01), Hacd3 (M-065373-01), Slc25a4 (M-061103-01), Slc25a5 (M-042392-01), Smpd4 (M-042323-01), Stt3b (M-062290-01), Surf4 (M-062783-00), and Tmem43 (M-046911-01) were purchased from Dharmacon. siRNA specific to Surf4 (Surf4 Stealth Select RNAi) purchased from Thermo Fisher Scientific. Negative control siRNA was purchased from Dharmacon and Thermo Fisher Scientific. A total of 20 nM siRNA was introduced to cells using Lipofectamine RNAiMAX (Invitrogen) according to the manufacturer's instruction. After 6 h, the medium was replaced by DMEM with 10% heat-inactivated FBS and cells were further incubated for 44 or 68 h for subsequent experiments.

**Statistics and reproducibility**. Error bars displayed throughout this study represent s.e.m. unless otherwise indicated, and were calculated from triplicate or quadruplicate samples. Statistical significance was determined with R (ver. 4.0.2) by one-way ANOVA followed by Tukey–Kramer post hoc test.; $*P < 0.05$; $**P < 0.01$; $***P < 0.001$; NS not significant ($P > 0.05$). Data shown are representative of three independent experiments (Figs. 1b; 2a–c; 3b–f; and 4a–c, e–g; Supplementary Figs. 1d, e; 2a–g; 4–7; 8a, b; 9–11; 12b; 13a–d; 14a, b; 15a, b, d; 16a, c; 17; and 19) and each yielding similar results.

**Reporting summary**. Further information on research design is available in the Nature Research Reporting Summary linked to this article.

## Data availability

The authors declare that the data supporting the findings of this study are available within the paper, Supplementary Information or from the corresponding author upon request. Source data are provided with this paper.

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

## Acknowledgements

This work was supported by JSPS KAKENHI Grant Numbers JP19H00974 (T.T.), JP15H05903 (T.T.), JP17H06164 (H.A.), JP17H06418 (H.A.), JP20H05307 (K.M.), JP20H03202 (K.M.), and JP17K15445 (K.M.); AMED-PRIME (17939604) (T.T.; Takeda Science Foundation (to S.W. and to K.M.), MSD Life Science Foundation (Public Interest Incorporated Foundation) (K.M.), Daiichi Sankyo Foundation of Life Science (K.M.), the Research Foundation For Pharmaceutical Sciences (K.M.), Young Investigator Grant (Graduate School of Life Sciences, Tohoku University) (K.M.), Foundation for Promotion of Cancer Research in Japan (K.M.), the Cell Science Research Foundation (K.M.), the Pharmacological Research Foundation Tokyo (K.M.), the Japan Foundation for Applied Enzymology (K.M.), and Tokyo Biochemical Research Foundation (K.M.), Center of Innovation program from Japan (K.M.), Grant for Basic Science Research Projects from the Sumitomo Foundation (K.M.), Koyanagi-Foundation (K.M.), and the Nakatomi Foundation (K.M.). We thank Atsuko Yabashi for her technical support in the immuno-EM.

## Author contributions

K.M. designed and performed the experiments, analyzed the data, interpreted the results, and wrote the paper; E.O. designed and performed experiments, analyzed data, and interpreted results; R.U. performed the experiments for identification of STING binding proteins; Y.K. performed the experiments with cGAS KO cells; F.K. constructed the plasmids; T.U. and S.W. performed the experiments with electron microscopy; T.S. and N.D. performed the proteomics analysis; H.A. designed the experiments, interpreted the results; A.K.S. discussed the results; T.T. designed the experiments, interpreted the results, and wrote the paper.

## Competing interests

The authors declare no competing interests.
