## [Peer Review File · Nature Communications]

REVIEWER COMMENTS

Reviewer #1 (Remarks to the Author):

This paper reports on the mechanism underlying an autoinflammatory disease called COPA syndrome. The authors show that loss-of-function mutations in COPA result in defective retrieval of the immune signaling protein STING from the Golgi—where STING signaling occurs—back to the ER. STING is normally activated to translocate to the Golgi and signal by cyclic-di-nucleotide (CDN) second messenger ligands. Interestingly, the authors show that retention of STING in the Golgi causes CDN-independent STING activation that requires palmitoylation of STING. The authors also report that retrieval of STING from the Golgi requires SURF4, which forms a complex with COPA and STING. Interestingly, CDN binding to STING or mutations in STING associated with a different STINGopathy (SAVI) disrupts the association of STING with COPA/SURF4. This latter result provides a long-sought explanation for how CDN binding leads to STING activation and also provides a molecular explanation for pathogenic SAVI mutations in STING. This is thus a highly important paper, not only for its identification of the cause of human inflammatory diseases, but also because of its fundamental insights into the mechanism of STING signaling. There are some issues with the experimental data, as described below, but assuming these can be addressed, I would favor publication in Nature Communications.

Figure 1. The effects in 1a appear relatively weak. I presume this is due to artifacts related to transient transfection as opposed to the much more dramatic effects seen with stable transfectants tested in 1b? Perhaps the authors can help the reader appreciate this, by acknowledging/explaining, e.g., “The effects of COPA on STING signaling were modest in this transient transfection system, possibly because of STING overexpression. Therefore, we examined the effects of wild-type or mutant COP in cells stably expressing STING in *Sting*^{-/-} MEFs”. As a related point, is it possible to compare the expression of STING and COP in the stable transfectants as compared to the endogenous levels of these proteins in WT MEFs? Lastly, I think some qRT-PCR data is needed to show that the cells expressing mutant COPA in 1b are inducing the transcription of IFN β as expected? My belief is that the effects on IFN expression should be much more dramatic than the reporter experiments in 1a.

Figure 2. The images in a and b need some kind of quantification, ideally by automated software or a blinded observer. Also, there should be some kind of indication of how representative the images shown are of the total population of cells (what fraction of the cells in the population show this phenotype? can a wider view image showing many cells also be shown so that we can get a sense of how representative the data are?). How was it determined that the vesicular or vacuolar compartments labeled with ‘G’ are in fact Golgi? At present, this is not convincing to me without some additional argument, e.g., a secondary gold label (of a different size) for a known Golgi marker?

Figure 3. For a, some quantification would help us to evaluate these data, e.g., an indication of the number of peptides observed for each protein? For e, can quantification of the several replicates of this experiment be shown and analyzed statistically?

Figure 4. For a, b, c, can qRT-PCR be shown to validate that the expected transcription of IFN-beta is observed?

Other points:

-COPA syndrome presents with different pathology than SAVI. The authors should perhaps comment on whether they think COPA is entirely explained by defective retrieval of STING from the Golgi, or whether other effects of COPA loss-of-function may contribute to the disease.

- SURF4 has previously been identified as a protein that mediates secretion of cargo (e.g., PCSK9 PMID 30251625, 29643117). This seems at odds with the proposal of the authors that SURF4 is actually involved in retrograde transport. Can this discrepancy be commented on? It is formally

possible that SURF4 knockdown somehow leads to accumulation of cytosolic DNA, leading to STING activation. Therefore, it would be nice to show that SURF4 knockdown still causes STING activation even in cGAS KO cells.

- Figure 1. The figure legend needs a description of the statistical tests. Ideally, the band intensities in the western blots in b would be quantified (across multiple replicates) and analysed statistically as well.

Reviewer #2 (Remarks to the Author):

Comments

The trafficking of STING from the ER to ERGIC/Golgi is required for its activation, which is essential for innate immunity to cytosolic DNA. Dysregulation of STING trafficking have been reported to be responsible for several autoimmune/ autoinflammatory diseases. In this manuscript, the authors found that STING was detained at the Golgi by mutation of the COPA gene. Furthermore, the authors found that the complex STING/Surf4/ α -COP retrieved STING back to the ER from the Golgi, which was disrupted by the disease-causative α -COP variants. Although the current findings are interesting and important, the lack of data from clinic samples or mouse models cripples current conclusions and importance of the study. In addition, this work conceptually overlaps with a preprint published by the same group in bioRxiv (doi:<https://doi.org/10.1101/2020.05.20.106500>) . Integrating this study with the findings in bioRxiv preprint will make the story more significant.

Other concerns are listed as below

1. The COPA gene encodes α -COP, one of the COP-I subunits. Do disease-causative α -COP variants affect assembly of the COP-I complex and lead to broad disruption of protein localization? It has been shown that the α -COP variants inhibit MAVS- and TRIF-mediated IRF activation, which seems to support this possibility. Another question is, do these variants disrupt STING retro-translocation by affecting other regulators of STING trafficking, eg. STIM1?
2. The authors need to clarify whether Surf4/ α -COP is responsible for retro-translocation of STING from the ERGIC to ER or from the Golgi to ER.
3. Phosphorylation of TBK1 depends on the trafficking and activation of STING. In Fig. 4 and Extended Data Fig. 9, the authors need to explain why palmitoylation inhibitors inhibited phosphorylation of TBK1 but not the ER-to-Golgi trafficking of STING in cells expressing the α -COP variants.
4. STING is localized at the ER in un-stimulated cells, is this ER localization of STING a result of homeostatic regulation by COP-I and COP-II? Does this mean that STING trafficking from the ER-to-Golgi also occurs in unstimulated cells?
5. Does attenuation of Surf4/ α -COP-mediated retro-translocation of STING affect ligand-dependent STING activation?

Reviewer #3 (Remarks to the Author):

This manuscript describes a new model for immune regulation via STING, whereby STING constitutively cycles between the ER and Golgi. Engagement with the COPI trafficking system, perhaps mediated by SURF4, functions to retain a steady state localisation for STING in the ER. This model is based on patient mutations in the COPI protein, alpha-COP, which cause STING to localize to the Golgi at steady state. The authors propose that pathogen activation of STING (via cGAS) triggers release from this COPI-retrieval cycle to result in acute localisation to the Golgi and immune signalling. Overall, this is an intriguing and well-executed study that will be of interest to the community. I have

several technical concerns and some textual clarifications that might improve the manuscript.

1. Where on the alpha-COP structure do the patient mutations lie? Do these destabilise the protein (presumably not from the westerns presented) or directly impact cargo binding? Some more explanation of these point mutations would be appreciated.

2. The alpha-COP expression experiments in HEK cells suggest that these effects are dominant negative since these cells presumably still have their endogenous alpha-COP. Or is the over expression condition sufficient to destabilize endogenous alpha-COP such that it is fully replaced by the mutant form? This is important to establish in interpreting the effects of the mutants.

3. In Figure 2C, the gold labelling should be quantified.

4. For the SURF4 siRNA experiments, rescue constructs should be tested to show specific effects of KD. This would also be a great opportunity to test the lysine retrieval mutants for an effect, which should mirror the KD situation. Similarly, the siRNA SURF4 experiments would be enhanced if the authors could demonstrate that co-IP between COPI and STING is reduced without SURF4. Again, this would suggest a direct effect rather than a potential indirect effect of perturbing ER-Golgi membrane flux.

5. The BFA experiments are confusing to me. BFA causes massive redistribution of the Golgi back to the ER, and this has classically been used to artificially activate proteins that must travel to the Golgi for processing (the best example of this is the SREBP cleavage upon sterol depletion). So, if the redistribution of Golgi resident processing enzymes to the ER fails to activate STING, then that suggests that there is another step required. Presumably this is binding of the ligand, which is missing in most of the conditions tested here. The authors might add DNA to show that then the BFA condition does in fact activate STING. Or otherwise clarify what they think is happening in this condition.

Reviewer #1 (Remarks to the Author):

This paper reports on the mechanism underlying an autoinflammatory disease called COPA syndrome. The authors show that loss-of-function mutations in COPA result in defective retrieval of the immune signaling protein STING from the Golgi—where STING signaling occurs—back to the ER. STING is normally activated to translocate to the Golgi and signal by cyclic-di-nucleotide (CDN) second messenger ligands. Interestingly, the authors show that retention of STING in the Golgi causes CDN-independent STING activation that requires palmitoylation of STING. The authors also report that retrieval of STING from the Golgi requires SURF4, which forms a complex with COPA and STING. Interestingly, CDN binding to STING or mutations in STING associated with a different STINGopathy (SAVI) disrupts the association of STING with COPA/SURF4. This latter result provides a long-sought explanation for how CDN binding leads to STING activation and also provides a molecular explanation for pathogenic SAVI mutations in STING. This is a thus a highly important paper, not only for its identification of the cause of human inflammatory diseases, but also because of its fundamental insights into the mechanism of STING signaling. There are some issues with the experimental data, as described below, but assuming these can be addressed, I would favor publication in Nature Communications.

Figure 1. The effects in 1a appear relatively weak. I presume this is due to artifacts related to transient transfection as opposed to the much more dramatic effects seen with stable transfectants tested in 1b? Perhaps the authors can help the reader appreciate this, by acknowledging/explaining, e.g., “The effects of COPA on STING signaling were modest in this transient transfection system, possibly because of STING overexpression. Therefore, we examined the effects of wild-type or mutant COP in cells stably expressing STING in *Sting*^{-/-} MEFs”. As a related point, is it possible to compare the expression of STING and COP in the stable transfectants as compared to the endogenous levels of these proteins in WT MEFs? Lastly, I think some qRT-PCR data is needed to show that the cells expressing mutant COPA in 1b are inducing the transcription of IFN β as expected? My belief is that the effects on IFN expression should be much more dramatic than the reporter experiments in 1a.

> Thank you for important suggestions.

(i) The following text has been included in the revised manuscript (page 5, line 11 - line 14). We cited the paper [Burdette et al., Nature 478, 515 (2011)], in which they carefully optimized the concentration of STING plasmid to examine the effect of cyclic dinucleotide.

“The effects of the α -COP variants on STING signalling were modest in this transient transfection system, possibly because of STING overexpression¹⁰. Therefore, we examined the effects of wild-type or mutant α -COP in cells stably expressing STING in *Sting*^{-/-} MEFs.”

(ii) We compared the expression level of EGFP-STING in [*Sting*^{-/-} MEF + EGFP-STING] with that of endogenous STING in WT MEFs and found that the amount of EGFP-STING was around 3-fold higher than that of endogenous STING. We included the result in the revised manuscript as Fig. S1d.

The molecular weight of endogenous α -COP and exogenous α -COP variants tagged with FLAG does not differ much, and thus they were not resolved in SDS-PAGE. We examined the amount of total " α -COP" (endogenous α -COP plus exogenous α -COP variant) by WB using anti- α -COP antibody and found that total amount of α -COP did not vary among wild-type MEFs and stable transfectants (Fig. S2f). These results indicated that the amount of exogenous α -COP was not high compared to that of endogenous α -COP. We also tried to estimate the amount of exogenous α -COP by a different approach. By tagging a bigger protein (mScarlet-I) than FLAG-tag, the exogenous α -COP (α -COP-mScarlet-I) could be resolved from endogenous α -COP. The new results in Fig. S2g indicated that the amount of exogenous α -COP was about 30% to that of endogenous α -COP.

(iii) We validated the level of interferon β mRNA in [*Sting*^{-/-} MEFs + EGFP-STING + α -COP] and found no significant upregulation.

Konno *et al.* recently reported that *Sting*^{-/-} MEFs that were reconstituted with the constitutively active STING variants (SAVI variants) did not show upregulation of interferon β mRNA [Figure 3A in Konno et al., Cell Rep 23, 1112 (2018)]. Instead, these cells showed the upregulation of a number of innate immunity genes (Figure 2C in the Cell Reports paper). Therefore, we examined these genes that were shown to be upregulated in the paper and found that 16 genes were upregulated in the disease-causative α -COP-expressing cells. These results were included in Fig. 1c and Fig. S3. The constitutive activation of STING in MEFs may result in the suppression of interferon β , but not the proinflammatory genes.

Figure 2. The images in a and b need some kind of quantification, ideally by automated software or a blinded observer. Also, there should be some kind of indication of how representative the images shown are of the total population of cells (what fraction of the cells in the population show this phenotype? can a wider view image showing many cells also be shown so that we can get a sense of how representative the data are?). How was it determined that the vesicular or vacuolar compartments labeled with ‘G’ are in fact Golgi? At present, this is not convincing to me without some additional argument, e.g., a secondary gold label (of a different size) for a known Golgi marker?

> Thank you for critical suggestions.

(i) The images in Fig. 2a was quantified with three subcellular localization categories - *i.e.*, "the Golgi", "ER and the Golgi", and "ER - by a blinded observer. The results showed that EGFP-STING mostly localized at the perinuclear Golgi in cells expressing the α -COP variants and were included in Fig. S18b.

(ii) The intensity of p-TBK1 in Fig. 2b was quantified by Image-J. The results showed the increased intensity of p-TBK1 per cell in cells expressing the α -COP variants and were included in Fig. S18c.

(iii) Wider images of Fig. 2a and 2b were included as Fig. S7 and S9.

(iv) We performed a double immunolabelling experiment (Fig. 2c and Fig. S19) with different sized colloidal golds to detect STING (12 nm) and α -COP (6 nm). The stacked structure with α -COP signals was assigned as the Golgi (Go). The gold labelling was quantified and the results were included in Supplementary Fig. 18d.

Figure 3. For a, some quantification would help us to evaluate these data, e.g., an indication of the number of peptides observed for each protein? For e, can quantification of the several replicates of this experiment be shown and analyzed statistically?

> Thank you for critical suggestions.

(i) We included the number of peptides that were identified by mass spectrometrical analysis in Fig. 3a.

(ii) We repeated the IP experiments four times and statistically analysed the data in Fig. 3e. The results in Fig. S18e suggested the significant reduction of the binding of mutant Surf4 to α -COP.

Figure 4. For a, b, c, can qRT-PCR be shown to validate that the expected transcription of IFN-beta is observed?

> As aforementioned, we found no upregulation of interferon β mRNA in [*Sting*^{-/-} MEFs + EGFP-STING + the α -COP variants]. Therefore, we examined several other genes (Ccl5, Cxcl10, and Apol9b) that were confirmed to be upregulated by the expression of the α -COP variants (Fig. 1c and Fig. S3). The results showed that the treatment with

BFA (Fig. 4a) and with C178 (Fig. 4b) abolished the expression of these genes (Fig. S15c). In contrast, cGAS KO (Fig. 4c) did not influence the expression of these genes as expected (Fig. S16b).

Other points:

-COPA syndrome presents with different pathology than SAVI. The authors should perhaps comment on whether they think COPA is entirely explained by defective retrieval of STING from the Golgi, or whether other effects of COPA loss-of-function may contribute to the disease.

> In the very recent publication [Deng et al., J Exp Med 217, e20201045 (2020)], a mouse model of COPA syndrome was developed. The COPA mice exhibited several phenotypes caused by immune dysregulation, which were essentially abolished in *STING^{gt/gt}* background. Therefore, we assume that the immune dysregulation in COPA syndrome would largely be explained by defective retrieval of STING to the ER. However, given the multiple cargo proteins carried by COP-I vesicles, other effects of α -COP loss-of-function may contribute to the COPA syndrome-specific symptoms, such as pulmonary hemorrhage. We commented on this issue in the text (page 11, line 3↑ - page 12, line 1).

"However, given the multiple cargo proteins transported by COP-I vesicles, other effects of α -COP loss-of-function may also contribute to the COPA syndrome-specific symptoms, which are not manifested in SAVI in which STING is constitutively active because of the STING mutations."

- SURF4 has previously been identified as a protein that mediates secretion of cargo (e.g., PCSK9 PMID 30251625, 29643117). This seems at odds with the proposal of the authors that SURF4 is actually involved in retrograde transport. Can this discrepancy be commented on? It is formally possible that SURF4 knockdown somehow leads to accumulation of cytosolic DNA, leading to STING activation. Therefore, it would be

nice to show that SURF4 knockdown still causes STING activation even in cGAS KO cells.

> Thank you for your comments.

(i) The paper [Emmer et al., *Elife* 7, e38839 (2018)] suggested that Surf4 functions as an ER cargo receptor mediating the efficient cellular secretion of PCSK9. Because Surf4 cycles between the ER and the Golgi, we assume that Surf4 may also function as a cargo receptor for the retrograde trafficking pathway during Surf4 is retrieved back to the ER. We cited the paper in the text (page 8, line 6 - line 9).

"Surf4, a multi-pass transmembrane protein, cycles among the ER, ERGIC, and the Golgi. Surf4 functions in the anterograde trafficking pathway from the ER¹⁹. ~~and~~ Surf4 is involved in membrane recruitment of COP-I²⁰, suggesting that Surf4 also functions in the retrograde trafficking pathway."

(ii) We knockdowned Surf4 in cGAS KO cells and found that STING lost the ER localization (Fig. S13a) and that TBK1 was phosphorylated (Fig. S13b). Therefore, we could rule out the possibility that Surf4 KD causes accumulation of cytosolic DNA.

- Figure 1. The figure legend needs a description of the statistical tests. Ideally, the band intensities in the western blots in b would be quantified (across multiple replicates) and analysed statistically as well.

>We added a description of the statistical tests to the legend to Fig. 1a. The band intensities in Fig. 1b were quantified across 5 independent experiments and the results were analyzed statistically (Fig. S18a). The results showed the significant increase of pTBK1 in cells expressing the α -COP variants.

Reviewer #2 (Remarks to the Author):

Comments

The trafficking of STING from the ER to ERGIC/Golgi is required for its activation, which is essential for innate immunity to cytosolic DNA. Dysregulation of STING trafficking have been reported to be responsible for several autoimmune/autoinflammatory diseases. In this manuscript, the authors found that STING was detained at the Golgi by mutation of the COPA gene. Furthermore, the authors found that the complex STING/Surf4/ α -COP retrieved STING back to the ER from the Golgi, which was disrupted by the disease-causative α -COP variants. Although the current findings are interesting and important, the lack of data from clinic samples or mouse models cripples current conclusions and importance of the study. In addition, this work conceptually overlaps with a preprint published by the same group in bioRxiv (doi:<https://doi.org/10.1101/2020.05.20.106500>) . Integrating this study with the findings in bioRxiv preprint will make the story more significant.

Other concerns are listed as below

1. The COPA gene encodes α -COP, one of the COP-I subunits. Do disease-causative α -COP variants affect assembly of the COP-I complex and lead to broad disruption of protein localization? It has been shown that the α -COP variants inhibit MAVS- and TRIF-mediated IRF activation, which seems to support this possibility. Another question is, do these variants disrupt STING retro-translocation by affecting other regulators of STING trafficking, eg. STIM1?

> Thank you for important suggestions.

(i) We examined the subcellular localization of other COP-I subunits (β , β' , δ , and γ) in cells expressing the α -COP variants, and found that they localized mainly at the Golgi as in wild-type MEFs (Fig. S2a-e). We also examined the expression levels of these COP-I subunits by WB, and found that the expression of the α -COP variants did not affect that of other COP-I subunits (Fig. S2f). Thus, the disease-causative α -COP variants appeared not to influence significantly the integrity and localization of the COP-I complex. In cells expressing the disease-causative α -COP variants, GM130 (a cis-Golgi protein), and TGN38 (a trans-Golgi network protein) localized at perinuclear

region as in wild-type MEFs (Fig. S2a-e, S5). In these cells, calreticulin and calnexin showed the reticular appearance throughout the cytoplasm as in wild-type MEFs (Fig. S4 and S15). These results suggested that the expression of disease-causative α -COP variants did not lead to a broad disruption of protein localization.

(ii) We examined the interaction of STING and STIM1 in MEFs, which we used throughout this study, by co-immunoprecipitation assay according to the paper [Srikanth et al., Nat Immunol 20, 152 (2019)] and did not find the interaction. So, at least in our MEFs, the tethering ability of STIM1 appears not to contribute to the ER localization of STING.

2. The authors need to clarify whether Surf4/ α -COP is responsible for retro-translocation of STING from the ERGIC to ER or from the Golgi to ER.

> Thank you for the critical comment. To address this issue, we examined the subcellular localization of Surf4. As shown in Fig. S14a, Surf4 localized at punctate structures, some of which were positive with α -COP (ERGIC and the Golgi), but not with TGN38 (the Golgi). Surf4 co-localized with ERGIC53 at these punctate structures (Fig. S14b), supporting that these puncta represent ERGIC. Based on these results, in particular the one that Surf4 co-localized with α -COP in ERGIC, but not in the Golgi, we assume that Surf4 is involved in the retrograde transport of STING from the ERGIC to the ER.

We commented on this issue in the text (page 9, line 3 - line 10).

“We examined the subcellular localization of Surf4. Surf4 localized at punctate structures, some of which were positive with α -COP, but not with TGN38 (Supplementary Fig. 14a), suggesting that these puncta represent ERGIC. Surf4 co-localized with ERGIC53 (a protein that circulates among the ER, ERGIC, and the Golgi) at these punctate structures (Supplementary Fig. 14b), supporting that these puncta indeed represent ERGIC. Based on these results, in particular the one that Surf4 co-localized with α -COP in ERGIC, we assume that Surf4 was involved in the retrograde transport of STING from the ERGIC to the ER.”

3. Phosphorylation of TBK1 depends on the trafficking and activation of STING. In Fig. 4 and Extended Data Fig. 9, the authors need to explain why palmitoylation inhibitors inhibited phosphorylation of TBK1 but not the ER-to-Golgi trafficking of STING in cells expressing the α -COP variants.

> As we reported previously [Mukai et al., Nat Commun 7, 11932 (2016)], palmitoylation was not required for STING trafficking from the ER to the Golgi (Fig. 2i and Fig. S9 in the paper). STING underwent palmitoylation at the Golgi and this process was required for TBK1/IRF3 activation by STING.

4. STING is localized at the ER in un-stimulated cells, is this ER localization of STING a result of homeostatic regulation by COP-I and COP-II? Does this mean that STING trafficking from the ER-to-Golgi also occurs in unstimulated cells?

> Thank you for important comments. To address this question, we knocked down Sar1a/Sar1b, which are two small GTPases responsible for the ER exit of STING [Ogawa et al., Biochem Biophys Res Commun 503, 138 (2018)]. cGAS KO cells were used to rule out any possible contribution of cytosolic DNA/cGAMP to STING stimulation. As shown in Fig. S17, depletion of Sar1a/Sar1b altered STING localization from the Golgi to the ER, in cGAS KO cells expressing the disease-causative α -COP variants. Therefore, these results clearly showed that STING constantly exits the ER in unstimulated conditions.

We commented on this issue in the text (page 10, line 9↑ - line 6↑).

“The translocation of STING from the ER to the Golgi in cGAS-knockout MEFs expressing the α -COP variants were suppressed by knockdown of Sar1a/b (Supplementary Fig. 17), which are two small GTPases responsible for the ER exit of STING¹³.”

5. Does attenuation of Surf4/ α -COP-mediated retro-translocation of STING affect ligand-dependent STING activation?

> To address this question, we stimulated cells expressing the disease-causative α -COP variants with DMXAA, a mouse STING agonist. As shown, the level of pTBK1 after stimulation with DMXAA was essentially the same among cells expressing wild-type or the disease-causative α -COP (Fig. S1e). Therefore, the results suggested that attenuation of Surf4/ α -COP-mediated retro-translocation of STING did not affect ligand-dependent STING activation.

This experiment also suggested that the STING signalling in cells expressing the disease-causative α -COP can be further activated two- to three-fold with the STING ligand.

Reviewer #3 (Remarks to the Author):

This manuscript describes a new model for immune regulation via STING, whereby STING constitutively cycles between the ER and Golgi. Engagement with the COPI trafficking system, perhaps mediated by SURF4, functions to retain a steady state localisation for STING in the ER. This model is based on patient mutations in the COPI protein, alpha-COP, which cause STING to localize to the Golgi at steady state. The authors propose that pathogen activation of STING (via cGAS) triggers release from this COPI-retrieval cycle to result in acute localisation to the Golgi and immune signalling. Overall, this is an intriguing and well-executed study that will be of interest to the community. I have several technical concerns and some textual clarifications that might improve the manuscript.

1. Where on the alpha-COP structure do the patient mutations lie? Do these destabilise the protein (presumably not from the westerns presented) or directly impact cargo binding? Some more explanation of these point mutations would be appreciated.

> Thank you for the comments. All of the mutations of the disease-causative α -COPs lie in the N-terminal WD40 domain [Watkin et al., Nat Genet 47, 654 (2015)]. The WD40 domain in α -COP has been implicated in the recognition of cargo proteins with di-lysine motif [Eugster et al., EMBO J 19, 3905 (2000)]. We revised Fig. S1 as reviewer3 suggested.

We examined the subcellular localization of other COP-I subunits (β , β' , δ , and γ) in cells expressing the α -COP variants, and found that they localized mainly at the Golgi as in wild-type MEFs (Fig. S2a-e). We also examined the expression levels of these COP-I subunits by WB, and found that the expression of the α -COP variants did not affect that of other COP-I subunits (Fig. S2f). Thus, the disease-causative α -COP variants appeared not to influence significantly the integrity and localization of the COP-I complex.

We included the following texts (page 3, line 6 - line 8 and page 5, line 7 \uparrow - line 1 \uparrow).

“All of the mutations¹ of the disease-causative α -COPs lie in the N-terminal WD40 domain (Supplementary Fig. 1a), which has been implicated in the recognition of cargo proteins⁵.”

“We then stably expressed wild-type or mutant α -COP in *Sting*^{-/-} MEFs expressing EGFP-STING. Both wild-type and mutant α -COP localized mainly at the Golgi (Supplementary Fig. 2a-e). Other COP-I subunits (β , β' , δ , and γ) also localized at the Golgi in these cells (Supplementary Fig. 2a-d). We also examined the expression levels of these COP-I subunits by western blot, and found that the stable expression of α -COP did not affect that of other COP-I subunits (Supplementary Fig. 2f, g).”

2. The alpha-COP expression experiments in HEK cells suggest that these effects are dominant negative since these cells presumably still have their endogenous alpha-COP. Or is the over expression condition sufficient to destabilize endogenous alpha-COP such that it is fully replaced by the mutant form? This is important to establish in interpreting the effects of the mutants.

> Thank you for critical comments. As mentioned above, we found that the localization and amount of other COP-I subunits (β , β' , δ , and γ) were not affected by the expression of the α -COP variants. As shown in Fig. S2g, the amount of exogenous α -COP was about 30% to that of endogenous α -COP. Given the character of the mutations, COP-I complex having the α -COP variants can be defective in cargo recognition, therefore the presence of the α -COP variants would decrease the number of functional COP-I complex. Therefore, as Reviewer3 pointed out, the effect of the disease-causative α -COP variants on STING trafficking and activation may be dominant negative.

We included the following texts (page 6, line 11 - line 16).

“Given that the α -COP variants can be defective in cargo recognition¹ (Supplementary Fig. 1a) and that the localization and amount of COP-I subunits including endogenous α -COP were not affected by the expression of the α -COP variants (Supplementary Fig. 2), the α -COP variants would decrease the number of functional COP-I complex.

Therefore, the effect of the disease-causative α -COP variants on activation of STING pathway may be dominant negative.”

3. In Figure 2C, the gold labelling should be quantified.

> We quantified the gold labelling and the results were included in Supplementary Fig. 18d and 19.

4. For the SURF4 siRNA experiments, rescue constructs should be tested to show specific effects of KD. This would also be a great opportunity to test the lysine retrieval mutants for an effect, which should mirror the KD situation. Similarly, the siRNA SURF4 experiments would be enhanced if the authors could demonstrate that co-IP between COPI and STING is reduced without SURF4. Again, this would suggest a direct effect rather than a potential indirect effect of perturbing ER-Golgi membrane flux.

>Thank you for important suggestions.

(i) We performed the KD/rescue experiments with wild-type Surf4 and the lysine-retrieval mutant (KKK>AAA). As shown in Fig. S13c, the expression of wild-type (siRNA-resistant) Surf4, but not the lysine-retrieval mutant of Surf4, rescued the ER localization of STING. These results suggested that the interaction of Surf4 with α -COP through the C-terminal KKK motif (Fig. 3e) is essential for the STING retrieval to the ER.

(ii) We examined the co-IP between α -COP and STING with or without Surf4. As shown in Fig. S13d, the co-IP was reduced by Surf4 KD. These results supported the notion that Surf4 bridges STING and α -COP.

5. The BFA experiments are confusing to me. BFA causes massive redistribution of the Golgi back to the ER, and this has classically been used to artificially activate proteins

that must travel to the Golgi for processing (the best example of this is the SREBP cleavage upon sterol depletion). So, if the redistribution of Golgi resident processing enzymes to the ER fails to activate STING, then that suggests that there is another step required. Presumably this is binding of the ligand, which is missing in most of the conditions tested here. The authors might add DNA to show that then the BFA condition does in fact activate STING. Or otherwise clarify what they think is happening in this condition.

> Thank you for important comments. In the previous publication [Mukai et al., Nat Commun 7, 11932 (2016)] , we found that STING underwent palmitoylation at the Golgi and activated TBK1 kinase at the trans-Golgi network. We also provided some evidence that the raft-lipid microdomain at the trans-Golgi network may function in clustering palmitoylated STING, thus facilitating autophosphorylation of TBK1. TGN is unique in having a different response to BFA treatment. Rather than redistributed to the ER, the majority of the TGN collapses around the microtubule organizing center [Reaves et al., J Cell Biol 116, 85 (1992)]. Therefore, BFA may not redistribute the components that are essential for STING activation at the TGN (e.g. palmitoylation enzymes and raft lipids) to the ER. The addition of dsDNA or DMXAA (a membrane-permeable STING agonist), did not activate STING in BFA-treated cells [Fig. S5d in Ishikawa et al., Nature 461, 788 (2009) ; Fig. 1d in Mukai et al., Nat Commun 7, 11932 (2016)], supporting this notion.

REVIEWERS' COMMENTS

Reviewer #1 (Remarks to the Author):

I am satisfied that the authors have responded appropriately to the previous round of reviews and that the manuscript is ready to be accepted.

Reviewer #2 (Remarks to the Author):

The authors did an excellent job answering the raised questions and the manuscript is much improved.

The only issues that need further discussion are:

1. According to the working model, STING is cycled between the ER and the ERGIC/Golgi in the absence of its ligand. What might be the trigger for STING's trafficking back to ERGIC/Golgi in unstimulated conditions?
2. Although STING is retrieved back to the ER, is it possible for STING to activate TBK1 during its transient stay in the ERGIC in unstimulated conditions? If the answer is yes, how is activated TBK1 regulated?

More discussion about these questions would be important.

Reviewer #3 (Remarks to the Author):

The revised manuscript satisfactorily addresses my concerns and the study is suitable for publication in Nature Comms. I do have one question that might be clarified textually (or may be a misunderstanding on my part). The authors state that STING activation in the COPA mutants is relatively modest in the transfection system, perhaps because of over expression of STING (p. 5). However, they also show that the COPA effects can be dominant negative, which suggests that impaired cargo binding in the context of these an additional COPA copy that functions normally saturates the retrieval pathway, leading to Golgi-retained STING. If this is true, then over expression of STING should saturate retrieval even in a wt situation, leading to heightened STING activation, not modest activation. Some additional clarity on the relationship between STING levels and retrieval saturation would help the reader understand the new model.

Reviewer #1 (Remarks to the Author):

I am satisfied that the authors have responded appropriately to the previous round of reviews and that the manuscript is ready to be accepted.

> We appreciated your comments for publication.

Reviewer #2 (Remarks to the Author):

The authors did an excellent job answering the raised questions and the manuscript is much improved.

The only issues that need further discussion are:

1. According to the working model, STING is cycled between the ER and the ERGIC/Golgi in the absence of its ligand. What might be the trigger for STING's trafficking back to ERGIC/Golgi in unstimulated conditions?
2. Although STING is retrieved back to the ER, is it possible for STING to activate TBK1 during its transient stay in the ERGIC in unstimulated conditions? If the answer is yes, how is activated TBK1 regulated?

More discussion about these questions would be important.

> We appreciated your comments for publication.

1. The protein export from the ER does not necessarily require receptor-mediated cargo recognition [Warren and Mellman, *Cell* **98**, 125-127 (1999)]. We assume that the ligand (cGAMP)-free STING may leak out constantly from the ER by “bulk flow” and reach ERGIC [Wieland et al., *Cell* **50**, 289-300 (1987)].

2. We found that phosphorylated TBK1, a hallmark of STING activation, was exclusively localized at the TGN (Fig. S8b) in the α -COP variants-expressing cells. Therefore, as we reported previously with ligand-bound STING [Mukai et al., *Nat*

Commun **7**, 11932 (2016)], ligand-free STING appears to activate TBK1 also at the TGN, not at the ERGIC. The COP-I-mediated retrograde transport from the ERGIC to the ER prevents ligand-free STING from reaching the TGN.

We commented on this issue in the text (page 10, line 10↑ - line 9↑).

“This exit may be with bulk flow as part of the membrane without association with COP-II coat subunits.”

Reviewer #3 (Remarks to the Author):

The revised manuscript satisfactorily addresses my concerns and the study is suitable for publication in Nature Comms. I do have one question that might be clarified textually (or may be a misunderstanding on my part). The authors state that STING activation in the COPA mutants is relatively modest in the transfection system, perhaps because of over expression of STING (p. 5). However, they also show that the COPA effects can be dominant negative, which suggests that impaired cargo binding in the context of these an additional COPA copy that functions normally saturates the retrieval pathway, leading to Golgi-retained STING. If this is true, then over expression of STING should saturate retrieval even in a wt situation, leading to heightened STING activation, not modest activation. Some additional clarity on the relationship between STING levels and retrieval saturation would help the reader understand the new model.

> Thank you for your comments. We agree with you in that overexpression of STING should saturate retrieval even in a wt α -COP situation, leading to heightened STING activation, not modest activation. Indeed, STING is originally identified using the overexpression system in HEK293T cells in the absence of its ligand [Ishikawa and Barber, *Nature* **455**, 674 (2008)].

In our experimental setup used in Fig. 1a, although we used HEK293T cells, we optimized the amount of the plasmids for transfection, so that we could see virtually no difference between control (the very left column) and STING only (the 7th column from the left).

We added some comments on this issue in the text (page 10, line 6 \uparrow - line 5 \uparrow).